# ST-Align: Multi-Scale Image–Gene Foundation Modeling for Spatial Transcriptomics via Spot–Niche Alignment

**Yuxiang Lin**[1]* **Ling Luo**[1]* **Ying Chen**[2]* **Xushi Zhang**[1], **Zihui Wang**[2], **Rongshan Yu**[1,2]†

[1]National Institute for Data Science in Health and Medicine, Xiamen University, Xiamen, China
[2]School of Informatics, Xiamen University, Xiamen, China
`linyuxiang@stu.xmu.edu.cn, luoling2001@stu.xmu.edu.cn, rsyu@xmu.edu.cn`

## Abstract

Spatial transcriptomics (ST) measures genome-wide gene expression together with tissue morphology at spatially indexed locations, enabling region-resolved molecular analysis that is not accessible to bulk sequencing or histology alone. Learning robust multimodal representations from ST is challenging because spot images are low resolution, spot-level gene vectors reflect mixed-cell composition, and biologically meaningful signal often depends on local neighborhoods rather than isolated spots.

We present **ST-Align**, a domain-adapted image–gene pretraining framework for ST that injects an explicit *spot–niche* inductive bias. ST-Align represents each spot together with a local neighborhood (niche) and aligns image and gene representations at three levels: spot-level image–gene alignment, niche-level alignment between neighborhood morphology and aggregated gene expression, and a cross-scale spot–niche objective that couples local and tissue-context signals.

We pretrain ST-Align on 1.3 million spot-level image–gene pairs from 573 curated human 10x Visium slides (STimage-1K4M) and evaluate (i) zero-shot transfer for spatial domain identification on six held-out human brain slices and (ii) image-to-gene prediction under patient-level splits. ST-Align improves spatial domain identification by **28.7%** over the best multimodal baseline (ARI 0.340 vs. 0.256) and reduces gene prediction error by **16.5%** (MSE 0.168 vs. 0.184), with particularly strong gains for non-laminar genes. Overall, these results support multi-scale spot–niche alignment as a useful design principle for ST representation learning in human 10x Visium data. Broader validation across tissues and ST technologies remains future work.

## 1 Introduction

Understanding how molecular programs are organized in space is fundamental to precision medicine and therapeutic discovery Tong et al. (2023); De Visser & Joyce (2023); Bejarano et al. (2021); Lin et al. (2026). Histopathology provides rich morphological signals, while transcriptomics provides molecular readouts; however, bulk RNA sequencing collapses heterogeneous tissue into a single profile and cannot localize programs to specific regions within a whole-slide image.

Spatial transcriptomics (ST) addresses this gap by measuring genome-wide gene expression at spatially indexed locations and pairing it with the corresponding histology Chen et al. (2015); Wang et al. (2018). Each spot typically aggregates transcripts from multiple cells (e.g., $\sim 55\,\mu\text{m}$ in 10x Visium), enabling spatially grounded analyses of tissue architecture, microenvironments, and disease mechanisms.

Recent work has explored adapting generic multimodal models (e.g., CLIP and PLIP) to ST-style image–gene learning Christensen et al. (2024); Sun et al. (2024); Huang et al. (2023). However,

---

*These authors contributed equally to this work.
†Corresponding author.

direct transfer is often suboptimal for ST because (i) many methods treat spots independently and underuse neighborhood context, (ii) spot image patches are extremely small and differ from natural image statistics, and (iii) mixed-cell spot gene profiles complicate alignment with local morphology.

We develop **ST-Align**, a domain-adapted image–gene pretraining framework for ST that explicitly models *spot–niche* structure. ST-Align learns alignment at three coupled levels: spot-level image–gene alignment, niche-level alignment between neighborhood morphology and aggregated gene expression, and cross-scale spot–niche alignment that ties local and tissue-context signals. We pretrain ST-Align on 1.3 million image–gene pairs from 573 curated human 10x Visium slides and evaluate it on two representative tasks: zero-shot spatial domain identification on six held-out human brain slices and image-to-gene prediction under patient-level splits. We view this setting as a controlled testbed for foundation-style ST pretraining, while noting that the present experiments do not by themselves establish broad cross-tissue or cross-technology generalization.

**Contributions.**

- **Multi-scale inductive bias for ST.** We formulate multimodal ST learning as coupled spot- and niche-level representation learning and introduce a cross-scale spot–niche contrastive objective.
- **Domain-adapted multimodal architecture.** We combine ST-specific adaptive encoders with an attention-based fusion network to learn aligned image–gene representations from spot- and niche-scale inputs.
- **Large-scale pretraining with controlled evaluation.** We pretrain on 1.3M ST spot pairs and demonstrate consistent gains on zero-shot spatial domain identification and image-to-gene prediction in held-out human 10x Visium settings, while explicitly characterizing broader cross-tissue and cross-technology validation as future work.

## 2 RELATED WORK

### 2.1 MULTIMODAL FOUNDATION MODELS IN PATHOLOGY

Large pathology image–text resources have enabled multimodal foundation models that improve representation learning and cross-modal retrieval. OpenPath and related datasets support adapting CLIP/PLIP-style objectives for pathology understanding and education Huang et al. (2023); Schuhmann et al. (2022); Yin et al. (2024); Li et al. (2024); Sun et al. (2025); Chen et al. (2025). These efforts primarily target image–text pairing rather than image–gene pairing and do not explicitly model the multi-scale structure unique to ST.

### 2.2 FOUNDATION MODELS FOR WSI AND TRANSCRIPTOMICS

**WSI foundation models.** Self-supervised learning has produced strong encoders for whole-slide images, including transformer-based approaches trained on large WSI corpora Wang et al. (2022a); Chen et al. (2024b); Xu et al. (2024a); Nechaev et al. (2024); Lu et al. (2024). These backbones are optimized for high-resolution tiles and may not transfer directly to extremely small ST spot patches.

**Transcriptomics foundation models.** Transcriptome foundation models are largely developed for single-cell RNA-seq, using reconstruction or masked modeling to capture gene expression structure Yang et al. (2022); Cui et al. (2024); Hao et al. (2024). ST spot profiles, however, reflect mixed-cell composition and spatial context, motivating ST-specific adaptation and alignment strategies.

### 2.3 IMAGE–GENE RESOURCES FOR SPATIAL TRANSCRIPTOMICS

Early multimodal resources often paired WSIs with bulk gene expression, producing one gene vector per slide. ST enables fine-grained pairing of local image patches and spot-level expression, supporting learning of image–gene representations at scale. Several databases and benchmarks curate ST across tissues and technologies, including CROST, SODB, STOmicsDB, and Aquila Wang et al. (2024); Yuan et al. (2023); Xu et al. (2024b); Zheng et al. (2023). STimage-1K4M Chen et al. (2024a) provides large-scale paired spot image–gene data suitable for foundation pretraining.

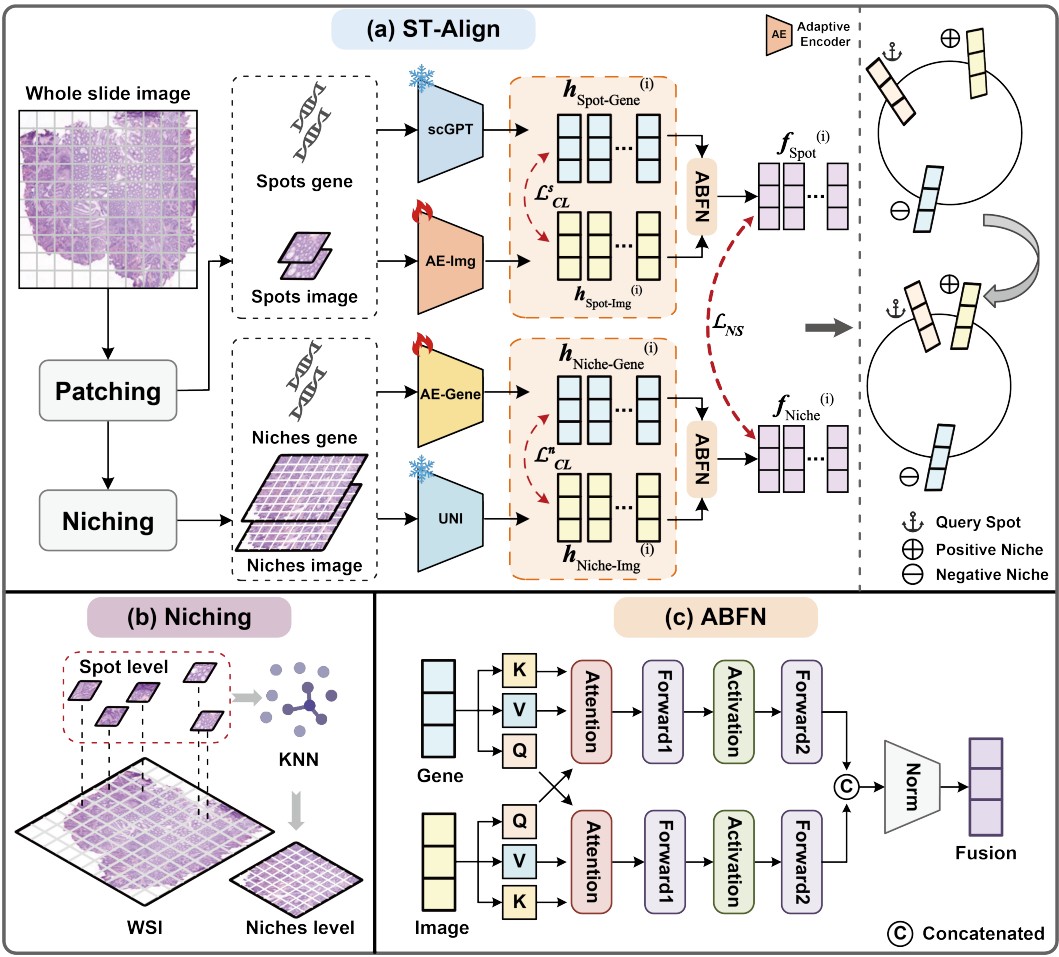

Figure 1: Overview of ST-Align. (a) Paired WSI and gene expression profiles are represented at both spot and niche scales. (b) Niche neighborhoods are constructed via $K$-nearest neighbors in spatial coordinate space. (c) An attention-based fusion network integrates image and gene features and is trained with multi-level contrastive objectives.

## 2.4 ST Tasks: Spatial Domain Identification and Gene Prediction

Learning embeddings that capture tissue organization is central to identifying spatial domains and microenvironments in ST. Spatially aware methods improve representation learning and clustering via graphs or neighborhood constraints Hu et al. (2021; 2024); Ma & Zhou (2024). A complementary goal is to model morphology–expression relationships for image-to-gene prediction or expression enhancement Zhang et al. (2024); Wang et al. (2022b); Si et al. (2024); Benjamin et al. (2024); Zhang et al. (2025). ST-Align targets both by learning aligned representations with explicit spot–niche structure.

## 3 Methods

We present **ST-Align**, a multi-scale image–gene pretraining framework for spatial transcriptomics (ST) (Figure 1). ST-Align models ST as a coupled *spot–niche* hierarchy, uses scale-aware encoders to address modality mismatch, and learns aligned representations through multi-level contrastive objectives. Our goal is to test whether an explicit spot–niche inductive bias improves representation learning in a controlled human 10x Visium setting; accordingly, we do not treat the present experiments alone as evidence of broad cross-platform generalization.

## 3.1 Multi-Level Spatial Representation

For slide $i$, let $X_i \in \mathbb{R}^{d_x \times d_y \times 3}$ denote the histology image and let $\{(\mathbf{c}_i^n, \mathbf{q}_i^n)\}_{n=1}^{N_i}$ denote the ST spots, where $\mathbf{c}_i^n \in \mathbb{R}^2$ is the spatial coordinate and $\mathbf{q}_i^n \in \mathbb{R}^{N_g}$ is the gene expression vector.

**Spot-level patches.** We crop a spot image patch $\mathbf{s}_i^n \in \mathbb{R}^{W_s \times H_s \times 3}$ centered at $\mathbf{c}_i^n$, yielding the set $S_i = \{\mathbf{s}_i^n\}_{n=1}^{N_i}$.

**Niche construction.** For each spot $n$, we form a niche neighborhood by selecting its $K$ nearest neighbors in coordinate space using Euclidean distance:

$$d(\mathbf{c}_i^n, \mathbf{c}_i^m) = \sqrt{\sum_{\ell=1}^{2} \left( c_{i,\ell}^n - c_{i,\ell}^m \right)^2}. \tag{1}$$

We set $K = 3$ as a conservative local neighborhood that captures immediate microenvironmental context while limiting oversmoothing across anatomical boundaries and keeping niche patches compact. We then crop a niche image patch $\mathbf{g}_i^n \in \mathbb{R}^{W_g \times H_g \times 3}$ as the bounding box covering the target spot and its neighbors (empirically ranging from $84 \times 84$ to $168 \times 168$ pixels), yielding $G_i = \{\mathbf{g}_i^n\}_{n=1}^{N_i}$. This value is fixed across all experiments rather than tuned per dataset; broader sensitivity analysis is left to future work.

**Niche-level gene expression.** We compute a niche gene vector by averaging spot expression over the neighborhood (including the target spot):

$$\mathbf{p}_i^n = \frac{1}{|\mathcal{N}_K(n)|} \sum_{j \in \mathcal{N}_K(n)} \mathbf{q}_i^j, \tag{2}$$

where $\mathcal{N}_K(n)$ denotes the neighborhood.

## 3.2 Encoders for ST Image–Gene Data

**Spot image encoder (adaptive).** Spot patches are extremely small (e.g., $28 \times 28$ pixels). We upsample them to $224 \times 224$ and encode them with a ResNet-50 trained on ST data:

$$\mathbf{r}_i^n = \text{AE-Img}(\mathbf{s}_i^n) \in \mathbb{R}^d. \tag{3}$$

**Niche image encoder (pretrained).** Niche patches contain larger tissue context. We encode niche images using a pretrained UNI encoder:

$$\mathbf{e}_i^n = \text{UNI}(\mathbf{g}_i^n) \in \mathbb{R}^d. \tag{4}$$

**Spot gene encoder (pretrained).** We encode spot-level gene expression using pretrained scGPT:

$$\mathbf{u}_i^n = \text{scGPT}(\mathbf{q}_i^n) \in \mathbb{R}^d. \tag{5}$$

We use scGPT as an expression prior rather than assuming that a Visium spot is equivalent to a single cell. Although each spot aggregates transcripts from multiple cells, it remains a structured vector in the same gene space, and the downstream ST-specific alignment and fusion modules are trained directly on spot-level data to absorb part of this distribution shift. We therefore treat scGPT as an initialization for representation learning in ST, while acknowledging that single-cell-to-spot mismatch remains a limitation.

**Niche gene encoder (adaptive).** We introduce an adaptive niche gene encoder (linear projection + Transformer encoder):

$$\mathbf{t}_i^n = \text{AE-Gene}(\mathbf{p}_i^n) \in \mathbb{R}^d. \tag{6}$$

### 3.3 ATTENTION-BASED FUSION NETWORK

Given image and gene embeddings at a given scale, we fuse them using symmetric cross-attention. Let $\mathbf{f}^I \in \mathbb{R}^d$ denote the image embedding and $\mathbf{f}^G \in \mathbb{R}^d$ denote the gene embedding. We compute:

$$\mathbf{z}^I = \text{softmax}\left(\frac{(\mathbf{f}^I W_q)(\mathbf{f}^G W_k)^\top}{\sqrt{d}}\right)(\mathbf{f}^G W_v), \tag{7}$$

$$\mathbf{z}^G = \text{softmax}\left(\frac{(\mathbf{f}^G W_q)(\mathbf{f}^I W_k)^\top}{\sqrt{d}}\right)(\mathbf{f}^I W_v). \tag{8}$$

The fused representation is:

$$\mathbf{h} = [\mathbf{z}^I W_I; \mathbf{z}^G W_G]. \tag{9}$$

### 3.4 MULTI-LEVEL ALIGNMENT OBJECTIVES

We train ST-Align with three contrastive objectives: spot-level alignment $\mathcal{L}^s_{CL}$, niche-level alignment $\mathcal{L}^n_{CL}$, and cross-scale niche–spot alignment $\mathcal{L}_{NS}$. For a minibatch of $B$ paired embeddings $\{(\mathbf{r}_i, \mathbf{u}_i)\}^B_{i=1}$, we use a symmetric InfoNCE loss:

$$\mathcal{L}^s_{CL} = -\frac{1}{2B}\sum_{i=1}^{B}\left[\log\frac{\exp(\mathbf{r}_i \cdot \mathbf{u}_i/\tau)}{\sum_{j=1}^{B}\exp(\mathbf{r}_i \cdot \mathbf{u}_j/\tau)} + \log\frac{\exp(\mathbf{u}_i \cdot \mathbf{r}_i/\tau)}{\sum_{j=1}^{B}\exp(\mathbf{u}_i \cdot \mathbf{r}_j/\tau)}\right], \tag{10}$$

where $\tau$ is a temperature. We apply the same formulation at the niche scale to obtain $\mathcal{L}^n_{CL}$. To couple scales, we align spot and niche fused features:

$$\mathcal{L}_{NS} = -\frac{1}{B}\sum_{i=1}^{B}\log\frac{\exp(\mathbf{h}^S_i \cdot \mathbf{h}^N_i/\tau)}{\sum_{j=1}^{B}\exp(\mathbf{h}^S_i \cdot \mathbf{h}^N_j/\tau)}. \tag{11}$$

We combine objectives as $\mathcal{L} = \lambda_1 \mathcal{L}^s_{CL} + \lambda_2 \mathcal{L}^n_{CL} + (1 - \lambda_1 - \lambda_2)\mathcal{L}_{NS}$ with $\lambda_1 = 0.4$ and $\lambda_2 = 0.3$. These coefficients are fixed across experiments to keep the spot-level, niche-level, and cross-scale terms on comparable scale while preserving a non-trivial contribution from $\mathcal{L}_{NS}$, rather than being tuned separately for each dataset. A full sensitivity analysis of $K$ and loss weights is left to future work.

## 4 EXPERIMENTS AND RESULTS

### 4.1 EXPERIMENTAL SETUP AND EVALUATION PROTOCOL

**Dataset composition:** All image–gene pairs are derived from STimage-1K4M Chen et al. (2024a), which spans multiple tissues and ST technologies. To reduce technical heterogeneity, we retain only human samples sequenced with 10x Visium and filter out WSIs with fewer than 50 spots. This yields 573 WSIs with 1.3 million spatially resolved spots.

**Data splitting:** We use two protocols. (1) *Spatial domain identification (zero-shot):* we evaluate on six human brain slices (151507–151673) from Maynard et al. (2021) that are completely held out during pretraining. (2) *Gene prediction:* we use patient-level splitting with an 80/20 train–test division, ensuring that spots from the same patient do not appear in both sets.

**Scope of claims:** Because both pretraining and downstream evaluation are restricted to curated human 10x Visium data, our results should be interpreted as evidence for strong within-platform representation transfer rather than as a claim of universal cross-tissue or cross-technology generalization.

**Implementation details:** ST-Align is implemented in PyTorch with distributed training across 3 NVIDIA A800 GPUs. AE-Gene uses a 6-layer Transformer encoder with 8-head attention and 0.1 dropout. Training uses AdamW with learning rate $5 \times 10^{-4}$, cosine scheduling, and weight decay in $[0.04, 0.4]$.

Table 1: **Spatial domain identification performance across foundation models.** Zero-shot clustering on six held-out human brain slices. G. and P. denote genomics and pathology modalities. Best in **bold**, second best underlined. Standard deviations computed over five runs.

| Model | Modality | | Brain Slice Dataset (ARI) | | | | | | Mean ARI |
|---|---|---|---|---|---|---|---|---|---|
| | G. | P. | 151507 | 151508 | 151509 | 151669 | 151670 | 151673 | |
| *Unimodal Pathology Models* | | | | | | | | | |
| CTransPath | | ✓ | 0.059±0.030 | 0.070±0.036 | 0.082±0.034 | 0.003±0.007 | 0.048±0.006 | 0.227±0.013 | 0.082 |
| UNI | | ✓ | 0.106±0.044 | 0.107±0.046 | 0.165±0.060 | 0.002±0.005 | 0.064±0.029 | 0.210±0.027 | 0.109 |
| Prov-GigaPath | | ✓ | 0.105±0.021 | 0.095±0.030 | 0.154±0.067 | 0.031±0.039 | 0.088±0.006 | 0.193±0.018 | 0.111 |
| Hibou | | ✓ | 0.067±0.033 | 0.061±0.034 | 0.075±0.040 | 0.013±0.029 | 0.086±0.003 | 0.220±0.010 | 0.087 |
| CONCH | | ✓ | 0.102±0.022 | 0.162±0.039 | 0.193±0.064 | 0.005±0.009 | 0.084±0.005 | 0.224±0.024 | 0.128 |
| *Unimodal Genomics Models* | | | | | | | | | |
| Scanpy | ✓ | | 0.218±0.031 | 0.225±0.018 | 0.390±0.026 | 0.288±0.202 | 0.233±0.164 | 0.229±0.027 | 0.264 |
| scFoundation | ✓ | | 0.206±0.021 | 0.233±0.021 | 0.387±0.027 | 0.285±0.061 | 0.259±0.060 | 0.199±0.031 | 0.262 |
| scGPT | ✓ | | 0.248±0.021 | 0.259±0.011 | 0.328±0.034 | 0.212±0.145 | 0.287±0.038 | 0.235±0.031 | 0.262 |
| *Multimodal Models* | | | | | | | | | |
| CLIP | ✓ | ✓ | 0.298±0.031 | 0.317±0.021 | 0.375±0.024 | 0.114±0.031 | 0.228±0.061 | 0.206±0.013 | 0.256 |
| PLIP | ✓ | ✓ | 0.271±0.040 | 0.301±0.008 | 0.421±0.018 | 0.092±0.051 | 0.179±0.036 | 0.227±0.012 | 0.248 |
| **ST-Align** | ✓ | ✓ | **0.310**±0.016 | **0.332**±0.035 | **0.470**±0.037 | **0.296**±0.100 | **0.352**±0.067 | **0.278**±0.014 | **0.340** |

## 4.2 Baseline Methods and Evaluation Metrics

**Pathology foundation models (P):** CTransPath Wang et al. (2022a), UNI Chen et al. (2024b), Prov-GigaPath Xu et al. (2024a), Hibou Nechaev et al. (2024), and CONCH Lu et al. (2024) are used as frozen encoders for spot-level image features.

**Genomics foundation models (G):** scFoundation Hao et al. (2024), scGPT Cui et al. (2024), and Scanpy Wolf et al. (2018) are used for transcriptomic feature extraction from ST spots.

**Multimodal baselines:** CLIP Radford et al. (2021) and PLIP Huang et al. (2023) are retrained following STimage-1K4M protocols, using FC layers to project features into 32-dimensional embeddings.

**Evaluation metrics:** We report Adjusted Rand Index (ARI; higher is better) for spatial domain identification and mean squared error (MSE; lower is better) for image-to-gene prediction.

## 4.3 Spatial Domain Identification Results

ST-Align achieves a mean ARI of 0.340, improving over the strongest multimodal baseline (CLIP: 0.256) by **28.7%**. Gains are consistent across slices, including challenging cases such as 151669. Genomics encoders outperform pathology-only encoders on this task, indicating that molecular profiles provide strong signals for domain identity. However, multimodal training yields complementary benefits: ST-Align improves over both unimodal and multimodal baselines. Figure 2 provides representative qualitative comparisons. Because this evaluation is restricted to six held-out human brain slices from the same platform, these results should be interpreted as evidence for strong within-platform zero-shot transfer rather than as evidence of universal generalization across tissues or ST technologies.

## 4.4 Gene Expression Prediction Analysis

ST-Align obtains the lowest overall MSE (0.168), a **16.5%** improvement over the best multimodal baseline (CLIP: 0.184). Category-wise, ST-Align shows especially strong gains for non-laminar genes, suggesting that niche context and multi-level alignment help recover signals that are not visually obvious at the spot level. We note that this benchmark covers nine representative genes rather than a transcriptome-wide evaluation, so the present result should be read as focused evidence that niche-aware alignment improves prediction on selected genes in this setting.

## 4.5 Ablation Study

Ablations in Table 3 indicate that each component contributes to performance. Adaptive encoders improve both tasks, highlighting the importance of ST-specific modeling. The attention-based fusion network outperforms simple concatenation, and the cross-scale niche–spot loss $\mathcal{L}_{NS}$ provides

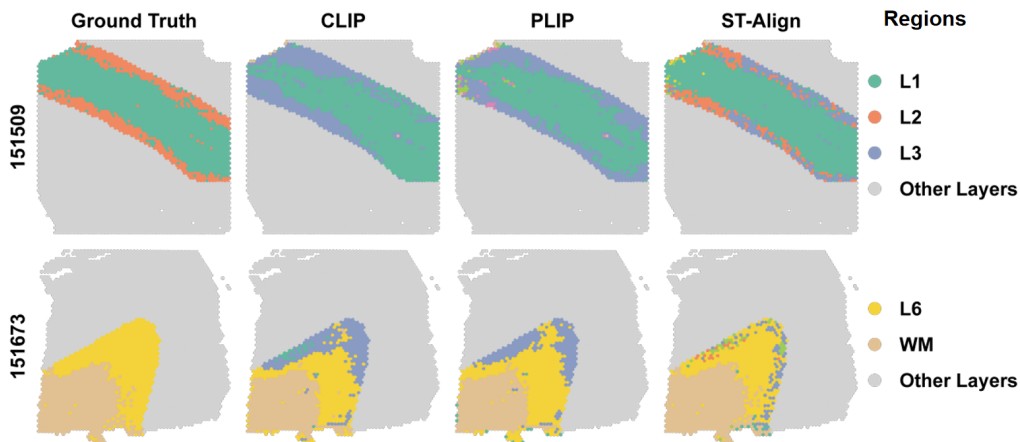

Figure 2: Qualitative spatial domain identification on representative held-out brain slices. ST-Align produces more coherent regions and cleaner boundaries than CLIP/PLIP.

Table 2: **Gene expression prediction performance by biological category.** Image-to-gene prediction accuracy (MSE; lower is better) across nine representative genes grouped into layer markers, laminar genes, and non-laminar genes.

| Model | Layer Marker Genes | | | Laminar Genes | | | Non-Laminar Genes | | | Overall MSE |
|---|---|---|---|---|---|---|---|---|---|---|
| | FABP7 | CCK | PVALB | PCP4 | MOBP | SNAP25 | IGKC | HBB | NPY | |
| *Unimodal Pathology Models* | | | | | | | | | | |
| CTransPath | $0.465_{\pm0.105}$ | $0.200_{\pm0.060}$ | $0.167_{\pm0.072}$ | $0.159_{\pm0.099}$ | $0.212_{\pm0.118}$ | $0.363_{\pm0.094}$ | $0.082_{\pm0.042}$ | $\underline{0.058}_{\pm0.026}$ | $0.032_{\pm0.034}$ | 0.193 |
| CONCH | $0.440_{\pm0.131}$ | $\mathbf{0.168}_{\pm0.069}$ | $0.175_{\pm0.067}$ | $0.189_{\pm0.122}$ | $0.222_{\pm0.151}$ | $0.347_{\pm0.091}$ | $0.067_{\pm0.033}$ | $0.089_{\pm0.048}$ | $\underline{0.027}_{\pm0.010}$ | 0.192 |
| Prov-GigaPath | $0.431_{\pm0.081}$ | $0.211_{\pm0.078}$ | $0.205_{\pm0.118}$ | $0.161_{\pm0.077}$ | $0.260_{\pm0.167}$ | $0.380_{\pm0.123}$ | $\underline{0.058}_{\pm0.014}$ | $0.072_{\pm0.024}$ | $0.039_{\pm0.047}$ | 0.202 |
| Hibou | $0.406_{\pm0.091}$ | $0.184_{\pm0.082}$ | $0.204_{\pm0.076}$ | $0.173_{\pm0.102}$ | $0.222_{\pm0.138}$ | $\underline{0.307}_{\pm0.085}$ | $0.075_{\pm0.021}$ | $0.066_{\pm0.031}$ | $0.028_{\pm0.008}$ | 0.185 |
| UNI | $0.478_{\pm0.101}$ | $0.194_{\pm0.049}$ | $0.182_{\pm0.056}$ | $0.151_{\pm0.088}$ | $0.283_{\pm0.200}$ | $0.383_{\pm0.070}$ | $0.069_{\pm0.045}$ | $\mathbf{0.049}_{\pm0.021}$ | $0.027_{\pm0.024}$ | 0.201 |
| *Multimodal Models* | | | | | | | | | | |
| CLIP | $\underline{0.394}_{\pm0.106}$ | $0.197_{\pm0.088}$ | $0.170_{\pm0.068}$ | $0.156_{\pm0.090}$ | $\underline{0.206}_{\pm0.083}$ | $0.321_{\pm0.112}$ | $0.076_{\pm0.038}$ | $0.112_{\pm0.030}$ | $0.034_{\pm0.040}$ | $\underline{0.184}$ |
| PLIP | $0.395_{\pm0.106}$ | $0.194_{\pm0.090}$ | $\underline{0.165}_{\pm0.069}$ | $\underline{0.150}_{\pm0.089}$ | $0.206_{\pm0.080}$ | $0.323_{\pm0.110}$ | $0.075_{\pm0.038}$ | $0.126_{\pm0.042}$ | $0.034_{\pm0.039}$ | 0.185 |
| **ST-Align** | $\mathbf{0.382}_{\pm0.075}$ | $\underline{0.175}_{\pm0.079}$ | $\mathbf{0.164}_{\pm0.087}$ | $\mathbf{0.148}_{\pm0.083}$ | $\mathbf{0.190}_{\pm0.113}$ | $\mathbf{0.298}_{\pm0.061}$ | $\mathbf{0.055}_{\pm0.031}$ | $0.074_{\pm0.022}$ | $\mathbf{0.027}_{\pm0.034}$ | $\mathbf{0.168}$ |

additional gains by coupling spot- and niche-scale representations. These results support the utility of the multi-scale formulation, although they do not by themselves constitute a full sensitivity analysis over neighborhood size or objective weights.

## 5 DISCUSSION

Spatial transcriptomics is inherently multi-scale: local molecular states are shaped by neighborhood context and tissue architecture. ST-Align explicitly models this structure by coupling spot- and niche-level representations and aligning them with a cross-scale objective, which yields consistent gains in zero-shot spatial domain identification and image-to-gene prediction.

**Why multi-scale alignment helps.** Spot images are extremely low resolution and noisy after resizing, while spot gene profiles reflect mixed-cell composition. Niche construction provides stabilizing context: neighborhood morphology offers more reliable tissue cues, and aggregated gene profiles reduce stochasticity. The ablation impact of the niche–spot loss $\mathcal{L}_{NS}$ is consistent with this interpretation.

**Limitations and future work.** This study intentionally focuses on curated human 10x Visium data to reduce confounding technical variation and isolate the contribution of spot–niche modeling. Accordingly, the present results support strong within-platform transfer, but they do not yet establish

Table 3: Ablation of key components. AEs: adaptive encoders; ABFN: attention-based fusion network; $\mathcal{L}_{NS}$: niche–spot contrastive loss.

| Configuration | ARI ↑ | MSE ↓ |
|---|---|---|
| UNI (image only) | 0.109 | 0.201 |
| scGPT (gene only) | 0.262 | – |
| Simple concatenation | 0.111 | 0.198 |
| ABFN + $\mathcal{L}_{NS}$ (w/o AEs) | 0.259 | 0.180 |
| AE + ABFN (w/o $\mathcal{L}_{NS}$) | 0.162 | 0.171 |
| **ST-Align (full)** | **0.340** | **0.168** |

cross-tissue, cross-species, or cross-technology generalization. In addition, niche construction uses a fixed $K = 3$ neighborhood with simple mean aggregation, and the objective weights are fixed rather than exhaustively tuned. Future work should test alternative neighborhood definitions, broader sensitivity analyses, and larger external validation sets spanning more tissues and ST platforms.

## 6 CONCLUSION

We presented **ST-Align**, a multi-scale image–gene pretraining framework for spatial transcriptomics that explicitly models spot–niche structure and learns alignment at spot, niche, and cross-scale levels. Pretrained on 1.3 million spot pairs from 573 human 10x Visium slides, ST-Align improves zero-shot spatial domain identification (mean ARI 0.340; +28.7% over the best multimodal baseline) and image-to-gene prediction (MSE 0.168; +16.5% improvement), with particularly strong gains for non-laminar genes. These results suggest that spot–niche alignment is a useful inductive bias for ST representation learning in human 10x Visium data and motivate broader validation across tissues and technologies in future work.

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
