# OpenReview forum: "ST-Align: Multi-Scale Image-Gene Foundation Modeling for Spatial Transcriptomics via Spot-Niche Alignment"
_ICLR.cc/2026/Workshop/FM4Science — ICLR 2026 Workshop FM4Science Poster_

### Official Review · Reviewer_rsGP · 2026-02-23
**ST-Align: Multi-Scale Image-Gene Foundation Modeling for Spatial Transcriptomics via Spot-Niche Alignment**

**Rating:** 5
**Confidence:** 3

**Review:**

Summary:

This paper presents ST-Align, a domain-adapted image–gene foundation model for spatial transcriptomics that injects an explicit spot–niche inductive bias. Each spot is paired with a local neighborhood (“niche”) built via KNN in coordinate space, and the model aligns image and gene representations at spot level, niche level, and cross-scale spot–niche level using multi-level contrastive objectives.

Pros:
1. The spot–niche formulation is a strong, ST-specific inductive bias that matches how spatial biology signal arises from local neighborhoods.
2. The method is clearly specified (niche construction, encoders, fusion, multi-level objectives) and the ablation supports that each component contributes.
3. Large-scale pretraining (1.3M pairs) and evaluation on both clustering and gene prediction provide credible evidence for representation quality.

Cons:
1. Absolute performance remains moderate (e.g., ARI 0.340), and the gains—while consistent—are not yet backed by analysis of when niche context helps most (e.g., varying K, missing neighborhoods, tissue types).
2. Domain identification is evaluated on six brain slices; broader transfer across tissues/technologies would strengthen the “foundation” claim.
3. Gene prediction is demonstrated on a small set of representative genes; a more comprehensive gene-set evaluation or additional downstream biological tasks would better validate utility.

---

### Official Review · Reviewer_o7TW · 2026-02-24
**Overall good, but some issues remain**

**Rating:** 6
**Confidence:** 2

**Review:**

# Summary
This paper proposes ST-Align, an adaptive image-gene model for the field of spatial transcriptomics (ST). This model introduces a spot-niche structure to perform multi-level contrastive learning and alignment of images and gene expression at the spot level, the niche level, and across scales. Overall, the paper's logic is sound, and the core idea of ​​multi-scale alignment (spot level, niche level, and cross-scale) is clearly conveyed. Explicitly decoupling and recombining spatial transcriptomics analysis into multi-level contrastive learning of "spot-niche" is an inspiring spatial inductive bias. This aligns more with biological intuition than simply throwing the entire image or isolated points into the model.

# Strengths
1. A spot-niche multi-scale spatial inductive bias that conforms to the characteristics of biological tissues is proposed, which effectively utilizes local neighborhood information.

2. It demonstrates outstanding performance improvements on fine-grained tasks such as non-laminar gene prediction.

# Weaknesses
1. You used a pre-trained scGPT at the spot level to encode gene expression. However, as you pointed out in the introduction, each spot (approximately 55 mM) in the 10x Visium technique contains mixed transcripts from multiple cells. scGPT is a model pre-trained on single-cell RNA-seq data. Applying a single-cell model directly to multi-cell (bulk-like) expression vectors lacking deconvolution processing requires an explanation as to why this approach theoretically does not generate a significant distribution shift.

2. The microenvironment is essentially an extremely complex high-order spatial topology. However, this paper constructs a niche by selecting $K=3$ nearest neighbors in Euclidean space and uses the most basic arithmetic mean (Equation 2) to aggregate gene expression in the microenvironment.

3. The paper claims to have pre-trained on a large-scale dataset (1.3 million spots), but for the zero-shot spatial domain recognition task, it was evaluated only on six datasets of the same type (held-out human brain slices). The core value of the base model lies in "generalization." Without zero-shot validation across tissue morphologies (e.g., migration from brain tissue to tumors, lymph nodes, or heart tissue), its claim as a "Foundation Model" cannot be supported. You must supplement it with at least 2-3 independent external validation sets of completely different tissue types.

4. You set $\lambda_1=0.4$ and $\lambda_2=0.3$ in the multi-scale contrastive learning objective. Please supplement the ablation experiments in Table 3 with the results of the grid search for these core hyperparameters to demonstrate that it is not overfitting to the current small test set.

5. The column headings in Figure 2 contain a clear spelling error, with "Regions" misspelled as "Reions". Such a basic mistake can severely damage a reviewer's first impression of the paper's rigor when submitting to top-tier conferences.

---

### Official Review · Reviewer_df8j · 2026-02-25
**Promising experimental results with limited technical novelty**

**Rating:** 6
**Confidence:** 2

**Review:**

This paper introduces ST-Align, a multimodal image-gene foundation model tailored to spatial transcriptomics (ST). The idea is to explicitly model spot-niche structure and align image and gene representations at three levels: spot-level, niche-level, and cross-scale spot-niche alignment. Although the problem setting and the model formulation are intuitive and well motivated, the architecture appears to be a heuristic combination of existing methods. For example, some design choices, such as the number of neighbors K=3 in Line 168 and specific weights of the loss terms in Line 239, are somewhat ad-hoc. It is unclear whether the reported gains stem from multi-scale alignment itself or from careful tuning. Additional ablation study and sensitivity analysis should be carried out.

On the other hand, the reported improvements over existing baselines are substantial, and the qualitative results in Figure 2 indeed show that ST-Align produces more coherent spatial domains. The ablation results, although limited, provide further understanding into how each component contributes to performance.

Overall, although there is limited novelty in methodology development, and there is no sensitivity analysis on hyperparameter choices, the empirical results are promising. This paper might be interesting to certain audience of the workshop.

---

### Official Review · Reviewer_P7Bd · 2026-02-25
**Review on ST-Align**

**Rating:** 7
**Confidence:** 3

**Review:**

This paper proposes ST-Align, a multimodal foundation model for spatial transcriptomics (ST) that introduces an explicit spot–niche multi-scale inductive bias. Unlike prior multimodal models that treat spatial spots independently, ST-Align models each spot jointly with its local neighborhood, aligning image and gene representations at multiple levels. The authors claim that the multi-scale alignment can better capture ST’s inherent spot-niche structure.

Strength:

•	The paper is well-motivated and grounded in the domain of spatial transcriptomics, as the proposed spot-niche framing is biologically relevant and aligned with the practice in spatial transcriptomics

•	The proposed multi-scale objective is conceptually strong and fits the biological problem structure

•	The model has been pretrained on large-scale real spatial transcriptomics data, which strengthens the experimental validity

•	 The experiments are relatively comprehensive and clean where the results show improvements for the proposed model and ablation table demonstrates incremental contributions of each module

Weakness:

•	There could be deeper analysis of the representation or attention map visualization, etc., to show how the multi-scale alignment could help

•	The generalization is questionable as the pretraining and evaluation are human only and on 10X Visium data

---

### Official Review · Reviewer_fVTs · 2026-02-25
**Multimodal and multiscale for gene spatial transcriptions and expression**

**Rating:** 6
**Confidence:** 3

**Review:**

This work proposes a multimodal and multiscale method to simultaneously address gene spatial transcriptions and gene expression.

**Strength**
- This model leverages existing models and simple attention schemes to combine them. This is a good and straightforward example of how specific models and datasets can be combined to produce a better model that addresses multiple tasks simultaneously while improving on all of them.
- This model significantly outperforms other benchmarks on the gene spatial transcription task.
- This model slightly outperforms other benchmarks in the gene expression task.

**Weaknesses**
- This paper does not seem complete, with many sections consisting of one or two sentences.
- Along the same lines, this work needs improvements to be more understandable. Currently, it is approachable by a general SciML audience, but it lacks sufficient explanations to be fully understandable.
- Context of how this model fits into a more general gene foundation model would be appreciated.

---

### Official Review · Reviewer_uT7P · 2026-02-26
**Reasonable method that exploits inductive bias in ST**

**Rating:** 6
**Confidence:** 4

**Review:**

Paper Summary: ST-Align learns aligned image-gene representations for spatial transcriptomics by modeling both individual spots and their local neighborhoods (niches). It combines pretrained encoders (UNI, scGPT) with adaptive components and trains three contrastive objectives at spot, niche, and cross-scale levels. Evaluated on zero-shot spatial domain identification (six brain slices) and image-to-gene prediction (nine genes).

Strengths:
1) The spot-niche idea makes sense. Individual Visium spots are tiny and noisy, so pulling in neighborhood context is a natural fix that generic models like CLIP/PLIP miss entirely. The paper explains this motivation clearly.

2) The ablation (Table 3) is the highlight. Removing the cross-scale loss L_NS causes the biggest drop, which directly supports the paper's main claim.

Weaknesses:
1) The evaluation is a bit narrow to call this work truly a "foundation model." Six brain slices from one dataset and nine genes falls a rather short to assess generalization. Even one non-brain tissue would help further this claim.

2) The niche is constructed by taking the K=3 nearest spatial neighbors for each spot, but the paper doesn't seem to ablate this choice or motivate why 3 is the right number.

Small Question:

This work restricts to 10x Visium, however, other ST technologies have quite different resolutions and spot sizes, so I'd be curious to see if the techniques employed would transfer well between ST methods.

---

### Meta-Review · Area_Chair_AmLy · 2026-02-27

**Recommendation:** Accept (Poster)
**Confidence:** 4

**Metareview:**

The average review score is above 6, which means reviewers recommended an acceptance.

---

### Decision · Program_Chairs · 2026-03-03

Accept (Poster)